# Sensor Technologies and Rehabilitation Strategies in Total Knee Arthroplasty: Current Landscape and Future Directions

**DOI:** 10.3390/s25154592

**Published:** 2025-07-24

**Authors:** Theodora Plavoukou, Spiridon Sotiropoulos, Eustathios Taraxidis, Dimitrios Stasinopoulos, George Georgoudis

**Affiliations:** 1Laboratory of Musculoskeletal Physiotherapy, Department of Physiotherapy, Faculty of Health and Caring Sciences, University of West Attica, Egaleo Campus, 12243 Athens, Greece; spiridonsotiropoulos@uniwa.gr (S.S.); taraxedes@gmail.com (E.T.); dstasinopoulos@uniwa.gr (D.S.); 2Department of Physiotherapy, University of West Attica, 12243 Athens, Greece; gg.physio@gmail.com

**Keywords:** Total Knee Arthroplasty (TKA), rehabilitation, wearable sensors, smart implants, mHealth, telerehabilitation, artificial intelligence, predictive analytics, remote monitoring

## Abstract

Total Knee Arthroplasty (TKA) is a well-established surgical intervention for the management of end-stage knee osteoarthritis. While the procedure is generally successful, postoperative rehabilitation remains a key determinant of long-term functional outcomes. Traditional rehabilitation protocols, particularly those requiring in-person clinical visits, often encounter limitations in accessibility, patient adherence, and personalization. In response, emerging sensor technologies have introduced innovative solutions to support and enhance recovery following TKA. This review provides a thematically organized synthesis of the current landscape and future directions of sensor-assisted rehabilitation in TKA. It examines four main categories of technologies: wearable sensors (e.g., IMUs, accelerometers, gyroscopes), smart implants, pressure-sensing systems, and mobile health (mHealth) platforms such as ReHub^®^ and BPMpathway. Evidence from recent randomized controlled trials and systematic reviews demonstrates their effectiveness in tracking mobility, monitoring range of motion (ROM), detecting gait anomalies, and delivering real-time feedback to both patients and clinicians. Despite these advances, several challenges persist, including measurement accuracy in unsupervised environments, the complexity of clinical data integration, and digital literacy gaps among older adults. Nevertheless, the integration of artificial intelligence (AI), predictive analytics, and remote rehabilitation tools is driving a shift toward more adaptive and individualized care models. This paper concludes that sensor-enhanced rehabilitation is no longer a future aspiration but an active transition toward a smarter, more accessible, and patient-centered paradigm in recovery after TKA.

## 1. Introduction

Total Knee Arthroplasty (TKA) is a widely accepted and increasingly common surgical intervention for patients with end-stage osteoarthritis or severe degenerative joint disease. Its primary objective is to alleviate pain, restore joint function, and improve quality of life. As the global aging population grows, the number of TKA procedures continues to rise, placing increasing emphasis on optimizing postoperative care and rehabilitation pathways to maximize patient outcomes.

Postoperative rehabilitation is critical to the success of TKA, influencing both the speed and quality of recovery. The aim of this review is to evaluate the current landscape of sensor-based rehabilitation technologies following TKA, and to identify key implementation challenges and opportunities for clinical integration. Traditional in-clinic rehabilitation programs, while effective, are resource-intensive and may not be feasible for all patients due to factors such as geographic distance, socioeconomic barriers, and limitations in mobility. Moreover, hospital-based rehabilitation alone often does not guarantee long-term functional success, as many patients struggle with adherence and engagement in prescribed exercise regimens at home [1]. This review is motivated by the increasing clinical need for scalable and personalized rehabilitation models, especially in resource-limited or remote settings.

In response to these challenges, the integration of sensor technologies into rehabilitation paradigms has emerged as a promising advancement. These technologies range from wearable motion tracking systems and inertial measurement units (IMUs) to smart implants, mobile health (mHealth) platforms, and AI-enhanced feedback tools. Wearable sensors are now capable of capturing real-time data on patients’ movement, gait patterns, and range of motion, offering valuable feedback for clinicians and patients alike [2].

In parallel, the rise of telerehabilitation platforms—such as ReHub^®^ and BPMpathway—has provided a feasible and effective way to extend care into the patient’s home, offering continuous engagement and monitoring without requiring their physical presence in a clinic [3]. These systems not only improve compliance but also reduce healthcare costs and enhance access for remote or underserved populations [2,4,5]. Home-based rehabilitation systems embedded with sensors have been shown to be safe and non-inferior to traditional models in terms of outcomes [6]. Additionally, intelligent systems powered by machine learning and real-time analytics have begun to redefine how patient recovery is assessed. For example, advanced platforms that incorporate algorithmic analysis of wearable sensor data have demonstrated high accuracy in predicting rehabilitation progression and customizing therapy plans accordingly [7]. These tools bridge the gap between clinician supervision and patient self-management, enabling hybrid models of care.

In addition, on the other hand, smart implant technologies are increasingly utilized intraoperatively and postoperatively. These devices provide surgeons with real-time feedback on ligament balance and alignment during surgery, potentially reducing complications such as arthrofibrosis [8]. App-controlled systems for neuromuscular electrical stimulation have also been employed to support muscle recovery and reduce rehabilitation costs [9]. Research continues to explore the impact of gamified rehabilitation platforms and portable systems that are both low-cost and highly engaging for patients. Such solutions can significantly improve adherence by increasing motivation and introducing real-time feedback mechanisms [10]. Systematic reviews further affirm the effectiveness of digital interventions, highlighting their diagnostic accuracy and potential to enhance rehabilitation outcomes [4,11,12,13].

In summary, the integration of sensors and intelligent monitoring systems into TKA rehabilitation represents a paradigm shift in postoperative care. These technologies address long-standing challenges in patient compliance, access, and personalization of therapy, thereby supporting a more holistic, data-driven, and patient-centered recovery model. A broad range of technologies has emerged, including joint-specific wearable devices, AI-enhanced feedback systems, and home-based remote monitoring platforms [14,15,16].

Nevertheless, sensor technologies alone are insufficient to guarantee optimal recovery outcomes. Their effectiveness depends heavily on integration within structured, individualized rehabilitation programs tailored to the clinical profile, physical limitations, and recovery goals of each patient [1,4]. Studies have shown that sensor-based interventions are most effective when used to complement—not substitute—clinician-guided therapy, particularly in cases where adherence, gait correction, and long-term engagement remain challenges [6,7,17]. Furthermore, successful implementation requires close collaboration among orthopedic surgeons, physiotherapists, biomedical engineers, and data analysts, forming a multidisciplinary care team capable of contextualizing sensor-derived metrics into meaningful clinical action [8,10,18]. Only through such collaborative, patient-centered frameworks can the full potential of digital rehabilitation tools be realized after TKA.

Despite growing interest in digital rehabilitation technologies, the literature remains fragmented regarding the classification, clinical integration, and comparative evaluation of sensor-based systems used in post-TKA recovery. Existing reviews often focus narrowly on specific sensor types or isolated rehabilitation stages, limiting their utility in guiding clinical practice [11,12]. This review aims to bridge that gap by synthesizing findings from recent studies involving wearable motion sensors, smart implants, and remote monitoring platforms, while critically analyzing their effectiveness, limitations, and usability in diverse rehabilitation settings [13,14,15,16,19]. By highlighting both technological advances and persistent implementation barriers, this narrative review seeks to offer a consolidated and forward-looking perspective for clinicians, researchers, and healthcare policymakers engaged in postoperative TKA rehabilitation.

## 2. Materials and Methods

This overview paper adopts a narrative review approach, aimed at summarizing and synthesizing current developments in sensor technologies applied to rehabilitation after Total Knee Arthroplasty (TKA). The selection of the literature was guided by a combination of structured and exploratory strategies.

A targeted literature search was conducted using the PubMed, Scopus, Web of Science, and Google Scholar databases, focusing on studies published between 2017 and 2024, to incorporate the most recent literature on a relatively new issue. Only studies published in English were considered.

The keywords included “Total Knee Arthroplasty,” “rehabilitation,” “wearable sensors,” “telerehabilitation,” “mHealth,” “smart implants,” and “artificial intelligence in physiotherapy”.

Inclusion criteria:Peer-reviewed original articles or systematic reviews.Studies involving sensor-based rehabilitation tools for TKA.Research that evaluated clinical, functional, or usability outcomes.

Exclusion criteria:Non-peer-reviewed sources (e.g., editorials, letters).Studies focusing solely on surgical techniques without postoperative rehab focus.

A flowchart of the study selection process is shown in Figure 1. The initial database search yielded 72 studies. After title and abstract screening, 26 full-text articles were assessed for eligibility, and 14 met the inclusion criteria for final analysis.

A total of 14 high-quality studies were selected for this review, comprising randomized controlled trials (RCTs), pilot studies, systematic reviews, and clinical validation research. The findings were categorized thematically across clinical applications, technology types, barriers, and future directions. A total of 14 studies were included in this review, covering diverse sensor-based technologies and study designs. A summary of the bibliographic and methodological characteristics of these studies is presented in Appendix A.

## 3. Challenges and Limitations in TKA Rehabilitation

Although Total Knee Arthroplasty (TKA) is a widely successful surgical intervention, its long-term outcomes are heavily influenced by the quality and consistency of postoperative rehabilitation. However, several systemic, technical, and behavioral challenges compromise the effectiveness of traditional rehabilitation protocols. These barriers are particularly evident in home-based recovery, which is becoming increasingly important as healthcare systems prioritize cost-effectiveness and outpatient care models.

### 3.1. Home Monitoring Limitations

A fundamental challenge in post-TKA rehabilitation is the limited monitoring capacity when patients are discharged from hospital settings. Without direct supervision, it becomes difficult to assess whether patients are performing exercises correctly or progressing as expected. Patients under hospital-based care have shown more consistent recovery metrics compared to those in unsupervised environments, largely due to the lack of real-time feedback and clinician oversight [11,20]. Similarly, although remote systems offer considerable promise, their implementation often lacks adequate support mechanisms for patients to report issues, leading to underreported complications or incorrect usage [5].

### 3.2. Variability in Patient Compliance

Patient adherence to rehabilitation programs is another persistent problem. Compliance varies due to multiple factors, including pain perception, motivation, cognitive function, and the ease of performing prescribed exercises. A randomized controlled trial demonstrated that even when telerehabilitation systems such as ReHub^®^ were deployed, consistent engagement remained a challenge for some patients, particularly older adults or those unfamiliar with digital tools [2]. Similarly, other studies have emphasized that home-based telerehabilitation systems are only as effective as the patient’s willingness and ability to follow structured routines [21]. These systems require intrinsic motivation, which may fluctuate due to psychosocial stressors or the absence of accountability mechanisms. Remote monitoring platforms enhanced by machine learning, while accurate in tracking data, have also shown significant variability in usage consistency among patients [8]. This highlights that even the most advanced tools require strong user engagement to yield meaningful outcomes

### 3.3. Lack of Individualized Therapy

Conventional rehabilitation often follows a one-size-fits-all model that does not account for inter-individual variability in healing, physical condition, or baseline function. This lack of personalization has been identified as a major limitation in optimizing recovery outcomes. Most digital rehabilitation interventions have been shown to fail in adjusting protocols based on dynamic patient feedback or sensor-based metrics, thereby limiting their effectiveness [7]. Randomized controlled trials have demonstrated that intelligent monitoring systems lead to superior outcomes, primarily due to their ability to adapt therapy in real time based on continuous data analysis [1]. Variability in gait entropy and peak frequency during treadmill recovery highlights the unique biomechanical recovery path of each patient, underlining the need for adaptive feedback models [3]. Further analysis of postoperative gait recovery has shown high individual variability in both motion complexity and peak frequency, indicating that motor control does not follow a uniform pattern across individuals [3]. Additionally, lateral body sway entropy varies substantially among patients, further supporting the value of dynamic, sensor-informed feedback systems [22]. Together, these findings provide compelling evidence for integrating adaptive feedback models in post-TKA rehabilitation to better align with each patient’s neuromuscular profile and optimize recovery outcomes.

### 3.4. Cognitive Load and Technology Literacy

An often-overlooked but significant barrier involves the cognitive demands and technology literacy required to operate some sensor-based or telehealth systems. Elderly populations, who constitute the majority of TKA recipients, have been shown to experience difficulty navigating platforms such as BPMpathway, despite their therapeutic potential [4]. This demographic-specific challenge must be carefully considered in the design of future solutions.

## 4. Sensor Technologies for TKA Rehabilitation

In recent years, the integration of sensor technologies into the rehabilitation process following Total Knee Arthroplasty (TKA) has revolutionized how clinicians monitor progress and deliver individualized care. These technologies offer objective, quantifiable insights into patient recovery, particularly in home-based and remote care settings. The four most prominent categories of such technologies are wearable sensors, smart implants, pressure-sensing systems, and mobile health (mHealth) platforms.

### 4.1. Wearable Sensors (IMUs, Accelerometers, Gyroscopes)

Wearable sensors represent one of the most widely studied and applied technologies in post-TKA rehabilitation. Inertial Measurement Units (IMUs), composed of accelerometers, gyroscopes, and sometimes magnetometers, are commonly attached to the lower limbs to assess joint angles, gait parameters, and overall mobility. These sensors enable high-resolution monitoring of range of motion (ROM), step count, and even biomechanical symmetry during walking.

Inertial measurement unit (IMU) sensors have been used to track changes in sample entropy and peak frequency during treadmill walking after TKA, demonstrating their value in capturing nuanced gait dynamics throughout recovery [3]. Similarly, wearable devices have been shown to effectively evaluate ROM progression, making them a viable alternative to clinic-based assessments [9]. Despite their clinical potential, challenges remain related to calibration, placement standardization, and data interpretation. Still, the technology has matured to a point where it can provide reliable metrics outside clinical settings.

### 4.2. Smart Implants

Smart implants are an emerging field in orthopedics, combining traditional prosthetic components with embedded sensors. These intraoperative sensors measure parameters such as joint alignment, ligament tension, and intra-articular pressure in real time, guiding surgeons to achieve optimal balance during TKA procedures. The use of electronic sensor devices during surgery has been shown to significantly reduce the incidence of arthrofibrosis, a common postoperative complication linked to improper ligament balancing [18]. Furthermore, smart implants continue to evolve, with biocompatible wireless components that allow postoperative monitoring without additional surgical intervention. Although promising, the application of such implants remains limited by high cost, surgical complexity, and limited long-term outcome data. Nonetheless, they offer a promising solution for high-risk or complex TKA cases.

### 4.3. Pressure-Sensing Insoles and Mats 

Pressure sensors embedded in insoles or pressure mats are another important category, particularly useful for assessing load distribution and gait asymmetries. These systems can measure plantar pressure profiles and ground reaction forces, which are critical for understanding compensatory movements and identifying abnormal gait patterns. While not as prominently featured in the recent clinical trials provided, such sensors have been integrated into machine learning-enhanced monitoring platforms to help identify trends in rehabilitation progress across diverse patient groups [3,5,15]. Such sensors are most often used in controlled research environments but are becoming increasingly accessible for clinical and home use. They are especially beneficial when paired with real-time feedback interfaces to guide patients during exercise sessions

### 4.4. Mobile Health (mHealth) and Smartphone-Based Tools

Mobile health platforms and smartphone apps are playing an increasingly critical role in enabling remote monitoring, coaching, and feedback for TKA patients. These solutions often integrate data from wearable sensors, allowing clinicians to track progress and modify rehab plans accordingly. The ReHub^®^ platform exemplifies how mHealth tools can support home-based rehabilitation with structured routines, video guidance, and performance analytics [3,5]. Similarly, home-based telerehabilitation systems have demonstrated feasibility with guided patient interfaces [11,14]. In more advanced applications, wearable and smartphone-connected platforms enhanced by machine learning have been validated to predict and monitor recovery trends in real time [9,15,16]. In conclusion, mobile systems offer scalable, cost-effective solutions and are especially valuable in rural or underserved areas. However, barriers such as internet access, digital literacy, and data privacy remain challenges to widespread adoption. Figure 2 shows a functional flowchart of sensor-based TKA rehabilitation, highlighting the role of digital tools and clinician interfaces.

To synthesize the evidence and provide a clearer understanding of the technologies applied in TKA rehabilitation, Table 1 presents a thematic classification of the 14 studies included in this review. Each study is categorized according to the type of sensor system used, its core functionality, and its primary clinical contribution.

## 5. Clinical Applications of Sensors

Sensor-based technologies have transformed the clinical landscape of postoperative rehabilitation for Total Knee Arthroplasty (TKA), enabling precise, real-time, and patient-centered care. By capturing kinematic and kinetic data, these systems allow clinicians to monitor and guide recovery in ways that were previously impossible with traditional subjective assessments. The main clinical applications of sensor technologies in TKA rehabilitation include mobility assessment, range of motion (ROM) monitoring, gait analysis, and real-time feedback delivery.

### 5.1. Mobility Assessment

One of the primary applications of wearable sensors is the assessment of mobility, especially during the early recovery phase. These sensors track spatiotemporal parameters such as step count, walking speed, cadence, and joint angles, which offer a reliable picture of a patient’s functional status. In recent MDPI studies, inertial sensors were used to track changes in gait complexity via sample entropy and peak frequency analysis during treadmill walking [3]. These advanced metrics revealed distinct recovery patterns that could not be captured through conventional observation alone. Additionally, intelligent monitoring systems have been shown to improve both mobility outcomes and therapy adherence by delivering real-time feedback on movement quality [1]. These systems are especially beneficial for detecting subtle deficits in mobility that may predispose patients to falls or asymmetrical gait, enabling proactive therapeutic interventions.

### 5.2. Monitoring Range of Motion (ROM)

Another crucial parameter for TKA recovery is the restoration of joint ROM, which is directly linked to a patient’s ability to return to daily activities. Traditionally measured using manual goniometers in clinical settings, ROM can now be tracked continuously and objectively through wearable or portable sensors. A sensor-based home rehabilitation system has been shown to accurately monitor improvements in knee flexion and extension angles. The system facilitated early detection of ROM plateaus and guided clinicians in adapting rehabilitation plans in real time [9,20]. Likewise, portable home systems have demonstrated that patients experienced consistent ROM gains that aligned closely with in-clinic standards [5]. These findings underscore the utility of sensor-driven ROM monitoring not only for convenience but also for improving clinical precision.

### 5.3. Gait Anomaly Detection

Gait abnormalities such as limping, asymmetry, or abnormal weight distribution are common after TKA and, if left unaddressed, can lead to long-term biomechanical dysfunctions. Sensor-enabled gait analysis allows clinicians to identify these problems objectively and early in the recovery process. A machine learning-based remote monitoring platform has been used to analyze wearable sensor data and detect deviations in gait patterns that correlated with delayed recovery outcomes [8]. Additionally, systematic reviews have emphasized that sensor-based assessments significantly improve diagnostic accuracy for gait dysfunctions when compared to traditional observation [7]. The ability to detect anomalies in real time can dramatically reduce recovery time and prevent compensatory movement patterns that hinder long-term joint health.

### 5.4. Real-Time Feedback to Patients

One of the greatest strengths of sensor-based systems is the delivery of real-time feedback, which is a powerful tool for improving patient adherence and motivation. Immediate insights into performance help patients self-correct and engage more meaningfully in their rehabilitation programs. Platforms such as ReHub^®^ offer structured sessions with real-time visual and auditory feedback to guide patients through exercises at home, leading to significantly improved adherence rates [2]. On the same lines, a home-based telerehabilitation platform equipped with real-time feedback has led to comparable, if not better, outcomes compared to traditional physical therapy sessions [21]. This immediate interactivity enhances patient autonomy and builds confidence, particularly valuable for individuals in remote or underserved areas.

## 6. Recent Research and Developments

Recent advancements in sensor-based rehabilitation for Total Knee Arthroplasty (TKA) reflect a dynamic and rapidly evolving field that merges orthopedic care with digital health, artificial intelligence, and wearable technologies. These innovations are aimed at enhancing the precision, efficiency, and personalization of rehabilitation protocols both in clinical settings and at home. One of the most notable trends is the proliferation of intelligent monitoring systems that leverage real-time sensor data to optimize recovery. A randomized controlled trial evaluating such a system showed significant improvements in mobility outcomes and rehabilitation efficiency. The system provided continuous tracking and adaptive feedback, highlighting the clinical potential of AI-assisted monitoring [1]. In the realm of telerehabilitation, an app-based digital program integrating real-time feedback and structured exercises was found to be both safe and effective compared to traditional physiotherapy. Patients using the platform showed comparable, and in some cases superior, functional recovery, underscoring the viability of remote interventions [2]. Parallel studies have validated the feasibility of home-based sensor systems. For example, telerehabilitation platforms tailored for elderly patients in diverse geographic contexts demonstrated excellent safety and patient satisfaction [4,21].

On the biomechanics front, inertial sensors have been applied to analyze treadmill gait patterns post-TKA, using advanced metrics such as entropy and frequency spectrum changes. This type of analysis not only deepens our understanding of biomechanical recovery but also enables early identification of abnormal gait trends [3]. Meanwhile, efforts to integrate machine learning into rehabilitation tools are gaining traction. A validated surveillance platform that combined wearable sensors with predictive analytics successfully tracked patient recovery, thus reducing the need for frequent in-person evaluations [8]. Overall, these developments are reshaping how TKA rehabilitation is delivered and monitored—shifting the focus from reactive, clinic-based care to proactive, data-driven, and remote strategies. As sensor technology becomes more affordable and interoperable, its adoption in routine orthopedic rehabilitation is expected to increase substantially. These challenges are summarized alongside practical implementation strategies in Table A1 (Appendix A).

## 7. Limitations and Barriers

Despite the promising advances in sensor-based rehabilitation technologies for Total Knee Arthroplasty (TKA), several critical limitations hinder their widespread clinical adoption and effectiveness. These barriers span technological, economic, demographic, and clinical domains.

### 7.1. Technological Limitations

One major challenge is the accuracy and consistency of sensor data. Variability in sensor placement, calibration, and movement artifacts can introduce errors in measurements, particularly in unsupervised home environments. The need for controlled conditions during gait assessments with inertial sensors has been emphasized, as entropy and frequency outputs are sensitive to environmental changes [3]. Moreover, many commercial platforms lack standardization in data interpretation, making cross-study comparisons and clinical integration difficult [7]. To address these challenges, platforms should adopt robust sensor calibration protocols and multi-sensor redundancy to minimize data errors. Moreover, applying machine learning techniques to filter movement noise can enhance data reliability [8]. Establishing standardized data collection and interpretation frameworks across manufacturers is essential for comparability and clinical scalability [7].

### 7.2. Cost and Accessibility

High costs remain a substantial barrier, especially for low-resource health systems and patients without insurance coverage. While some tools like BPMpathway or ReHub^®^ have shown cost-effectiveness in clinical trials, initial implementation expenses—such as sensors, software licenses, and training—can be prohibitive for smaller clinics [2,4]. Additionally, smart implants, though clinically promising, are still limited to niche applications due to their complexity and financial cost [18]. To mitigate these technological and cost-related limitations, platforms should invest in affordable, well-calibrated sensor hardware and promote interoperability across systems. Leveraging machine learning algorithms to filter signal noise may also improve data accuracy [8], while standardized frameworks for data interpretation and reporting are crucial to support clinical integration and cross-study comparability [7].

### 7.3. Elderly Population and Digital Literacy

Many TKA patients are elderly and may struggle with digital interfaces and device handling. A portion of patients required frequent technical support, reducing the independence that remote rehab platforms aim to foster [21]. Similarly, cognitive overload and low health-tech literacy remain major roadblocks for the widespread adoption of intelligent systems [6]. Developers should focus on intuitive user interfaces, voice command functionalities, and involving caregivers through remote coaching modules. Structured training sessions for patients and caregivers, along with accessible help desks, can significantly reduce digital dropout rates [6,21].

### 7.4. Data Interpretation and Clinical Integration

A final but crucial barrier is the interpretation and integration of high-volume sensor data into clinical workflows. Clinicians often lack the training or time to analyze raw sensor outputs or to act on AI-generated recommendations. Despite the potential of machine learning-powered surveillance platforms, success depends on clinician trust and seamless data integration with electronic health records [8]. Addressing this challenge requires training programs for clinicians, integrating decision-support dashboards, and embedding automated alerts in EHR systems. Facilitating collaboration between data scientists and clinicians during platform development can ensure usability and clinical relevance [8].

## 8. Future Directions

The integration of advanced sensor technologies in Total Knee Arthroplasty (TKA) rehabilitation marks a paradigm shift in orthopedic care. As current systems evolve, future developments are expected to move toward more intelligent, individualized, and remotely accessible solutions that will empower both clinicians and patients.

### AI-Based Personalized Rehabilitation

The findings of this review clearly demonstrate that recovery following Total Knee Arthroplasty (TKA) follows highly individualized trajectories, with significant variability in both gait patterns and patient adherence. Studies have revealed substantial variability in indicators like gait entropy and peak frequency, reinforcing the need for dynamic and personalized feedback models [3]. Intelligent monitoring systems that adapt therapy in real time based on continuous data analysis result in significantly better functional outcomes compared to static rehabilitation protocols [1].

In this context, artificial intelligence (AI) emerges as the next pivotal step. The ability of AI-enhanced systems to process sensor-derived data in real time, predict recovery trajectories, and automatically adjust therapy creates the foundation for truly individualized rehabilitation. A platform that integrated wearable sensors with machine learning algorithms serves as an early example of this trend, supporting remote patient monitoring while reducing the need for frequent in-person visits [8].

Recent applications of artificial intelligence in TKA rehabilitation primarily utilize adapted and fine-tuned versions of existing machine learning algorithms rather than entirely novel frameworks. These include support vector machines (SVMs), decision trees, and neural networks trained on gait patterns, joint angles, and patient adherence metrics. For instance, a supervised learning model was implemented to track compliance and flag deviations in recovery [8], while other systems have used convolutional neural networks (CNNs) for motion recognition tasks [17]. Although these models have demonstrated promising accuracy, future directions may focus on integrating multimodal data streams to develop context-aware predictive models for individualized rehabilitation guidance [20].

As these technologies continue to mature, the future of post-TKA rehabilitation is shifting toward smart, interoperable, and remote solutions that can dynamically adapt to the needs of each patient. Achieving this vision will require cross-disciplinary collaboration among engineers, clinicians, and data scientists to develop platforms that are technically robust, clinically meaningful, and human-centered.

## 9. Conclusions

Sensor technologies have redefined the framework of postoperative recovery following Total Knee Arthroplasty (TKA), providing real-time, objective data that enhance both clinical decision-making and patient autonomy. From wearable IMUs and smart implants to mHealth platforms and AI-powered systems, these tools address persistent challenges in conventional rehabilitation—such as limited supervision, inconsistent adherence, and lack of personalization.

Contemporary evidence confirms that such technologies can reliably monitor mobility, assess range of motion, detect gait anomalies, and deliver immediate feedback. Particularly, studies highlight the clinical value of metrics like gait entropy and peak frequency in capturing the individualized nature of recovery trajectories. Despite remaining limitations—including technical complexity, digital literacy disparities, and barriers to integration—sensor-assisted rehabilitation now represents a validated, patient-centered alternative to traditional models.

By bridging the gap between clinical oversight and self-managed care, sensor-based systems are shaping a more flexible, accessible, and data-driven rehabilitation model. Their increasing adoption affirms their role not merely as supplementary innovations but as fundamental tools in the advancement of modern rehabilitation practice.

However, sensor-based approaches must be integrated within comprehensive rehabilitation protocols that consider the broader clinical picture. Personalized rehabilitation remains a cornerstone for successful outcomes, and sensor systems should be regarded as augmentative tools—enhancing, but not replacing, expert clinical judgment and interdisciplinary care. Only through such synergy can the full potential of technology-assisted rehabilitation be realized following TKA.

## Figures and Tables

**Figure 1 sensors-25-04592-f001:**
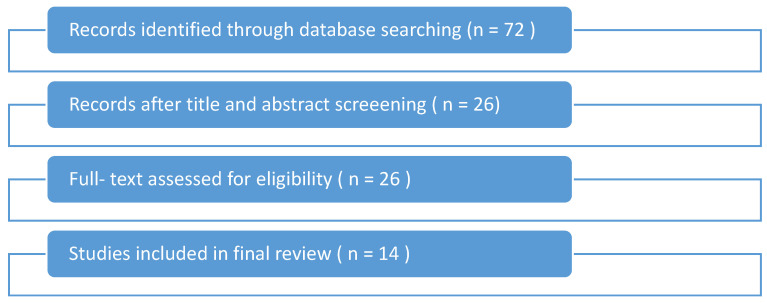
Flowchart of the study selection process.

**Figure 2 sensors-25-04592-f002:**
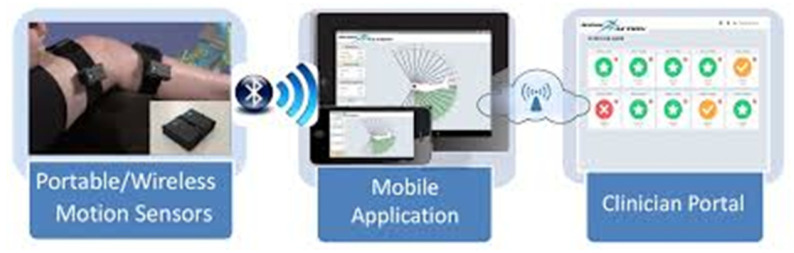
Functional flowchart of sensor-based rehabilitation after Total Knee Arthroplasty (TKA), integrating wearable sensors, smart implants, mHealth platforms, and clinician feedback loops.

**Table 1 sensors-25-04592-t001:** Classification and Characteristics of Reviewed Studies.

Study	Country	Year	Technology Category	Specific Tool or System	Participants	Study Design	Functionality	Key Finding	Key Outcomes
Xie et al. (2024) [1]	China	2024	AI monitoring	Intelligent system	60 TKA patients	RCT	Personalized feedback	Functional outcomes	Faster ROM recovery, improved feedback
Nuevo et al. (2024) [2]	Spain	2024	mHealth	ReHub platform	80 patients	RCT	Telerehabilitation	Non-inferior to in-clinic	Safe, non-inferior to clinic
van de Ven et al. (2023) [3]	Netherlands	2023	Wearables	IMUs	20 subjects	Observational	Gait entropy/frequency	Detected recovery patterns	Captured gait entropy changes
Msayib et al. (2017) [6]	UK	2017	Remote monitoring	Sensor network	25 patients	Feasibility	Home-based tracking	Feasible, safe	Safe and reliable home monitoring
Raje et al. (2024) [7]	India	2024	Review	Multiple tools	-	Systematic review	Systematic analysis	Highlighted digital gaps	Gaps and trends in digital rehab
Ramkumar et al. (2019) [8]	USA	2019	Wearables + ML	Predictive system	40 patients	Validation Study	Rehab prediction	Accurate, remote monitoring	Accurate recovery prediction
Huang et al. (2020) [9]	Taiwan	2020	ROM sensors	Knee ROM device	30 patients	Pilot study	ROM tracking	Clinic-equivalent accuracy	Clinic-equivalent ROM tracking
Pua et al. (2024) [10]	Singapore	2024	mHealth	Telemonitoring	100 patients	RCT	Exercise adherence	Non-inferior outcomes	Non-inferior outcomes
Hong et al. (2024) [4]	China	2024	mHealth	BPMpathway	65 patients	Prospective	Home rehab interface	Effective, literacy issues	Effective, some literacy barriers
Liptak et al. (2019) [20]	Australia	2019	Rehab devices	Maxm Skate	Protocol	RCT protocol	Strength training	Study protocol only	Strength training study proposed
Bell et al. (2020) [5]	USA	2020	Portable systems	Home rehab unit	28 patients	Pilot RCT	Remote ROM tracking	Matched clinical rehab	Matched clinical rehab outcomes
Salehian et al. (2024) [21]	Iran	2024	Telerehabilitation	Home device	40 patients	Controlled trial	Prescribed exercise	Safe, effective	Safe, effective recovery
Chughtai et al. (2017) [17]	USA	2017	App-controlled device	NMES system	50 users	Device study	Neuromuscular stimulation	Self-managed use	Self-management support
Geller et al. (2017) [18]	USA	2017	Smart implants	Ligament sensor	100 surgeries	Cohort study	Intraoperative alignment	Arthrofibrosis	Arthrofibrosis rate

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
