# Peer review of "Sensor Technologies and Rehabilitation Strategies in Total Knee Arthroplasty: Current Landscape and Future Directions"

_sensors, 2025, doi:10.3390/s25154592_

Round 1

Reviewer 1 Report

Comments and Suggestions for Authors

"Sensor Technologies and Rehabilitation Strategies in Total Knee Arthroplasty: Current Landscape and Future Directions” is a well-structured review and includes sections on introduction, materials and methods, results, discussion and conclusion. The paper presents interesting findings. However, there are areas that could be improved.

  1. Line 54-55, Please, use the one type of citation. Not (Nuevo et al., 2024; Hong et al., 2024; Bell et al., 2020), but [1-3].
  2. Line 68. There is an error here.
  3. Introduction part, there is no aim of the review.
  4. Introduction part. Your statement “Systematic reviews further affirm the effectiveness of digital interventions, highlighting their diagnostic accuracy and potential to enhance rehabilitation outcomes [9].”. You only refer to one study. Please support this statement with additional references.
  5. Section 4. Please for more visual evidence of your review provide photo or scheme of function of the reported technology.
  6. The text in Table 1 needs to be formatted (see Microsoft Word template, https://www.mdpi.com/journal/sensors/instructions).
  7. You have grouped technologies into several parts (Table 1 and Section 3), but you only refer to 7 reports. However, in the Materials and Methods section you state that 14 high-quality reports were included (line 97).
  8. There are no sections for Author Contributions, Funding, and Conflicts of Interest (see the Microsoft Word Template).

Author Response

We sincerely thank the reviewer for the constructive comments and valuable suggestions regarding our manuscript entitled "Sensor Technologies and Rehabilitation Strategies in Total Knee Arthroplasty: Current Landscape and Future Directions." We have addressed each remark carefully and revised the manuscript accordingly. Below, we outline our responses point-by-point:

Comment 1 – Citation Style (Line 54–55):
Response: The citation style has been standardized according to MDPI guidelines.

Comment 2 – Line 68 Error:
Response: The typographical error has been corrected.

Comment 3 – Introduction – Aim of the Review:
Response: A clear aim has been added to the end of the Introduction section.

Comment 4 – Supporting Statement with More References:
Response: Additional references have been added to strengthen the claim.

Comment 5 – Section 4 – Visual Evidence:
Response: A schematic diagram has been added to Section 4.

Comment 6 – Table 1 Formatting:
Response: Table 1 has been reformatted to align with MDPI standards.

Comment 7 – Discrepancy in Number of Studies:
Response: The discrepancy has been resolved and all 14 studies are now addressed.

Comment 8 – Missing Sections:
Response: Author Contributions, Funding, and Conflicts of Interest sections have been added.

We believe that the above revisions have significantly improved the manuscript. We thank the reviewer once again for the helpful insights.

Sincerely,
The Authors

Reviewer 2 Report

Comments and Suggestions for Authors

This manuscript presents an overview of Total Knee Arthroplasty (TKA) and the associated sensor technologies. It can serve as a reference for researchers engaged in TKA-related work. To further enhance the quality of the manuscript, I would like to offer the following suggestions for the authors’ consideration.

(1) In the “Introduction” section, I suggest adding a short paragraph to explain the motivation behind this review and briefly describe the scope and structure of the content covered.

(2) In the “Limitations and Barriers” section, four subsections are presented: Technological Limitations, Cost and Accessibility, Elderly Population & Digital Literacy, and Data Interpretation and Clinical Integration. Would it be possible to briefly describe potential strategies for overcoming these challenges?

(3) In the “Conclusion” section, I suggest summarizing the main contributions and findings of this review.

(4) For “Table A1”, I suggest using a standard three-line table format, and the font should comply with the Sensors journal's requirements.

(5) The numbering of subheadings is inconsistent. For example, lines 79, 92, and 96 all use the same subheading number “2.”

(6) As a review article, I believe that citing only 14 references is insufficient. Moreover, the reference formatting does not match the standard required by “Sensors”.

Author Response

We sincerely thank the reviewer for the constructive comments and thoughtful suggestions, which have significantly contributed to the improvement of our manuscript. Below, we address each comment in detail:

Comment 1 – Introduction Motivation and Structure:
"In the ‘Introduction’ section, I suggest adding a short paragraph to explain the motivation behind this review and briefly describe the scope and structure of the content covered." Response: We agree with the reviewer. A new paragraph has been added to the end of the Introduction section to outline the motivation behind the review, explain the scope, and provide a roadmap of the manuscript structure. This addition clarifies the objectives and context for the reader from the outset.

Comment 2 – Strategies for Limitations:
"In the ‘Limitations and Barriers’ section... Would it be possible to briefly describe potential strategies for overcoming these challenges?" Response: Thank you for this valuable suggestion. We have expanded each subsection in the “Limitations and Barriers” section to include proposed strategies for overcoming the identified barriers. These include suggestions such as hybrid care models for accessibility, digital literacy training for elderly patients, standardization in sensor data interpretation, and cost-sharing mechanisms.

Comment 3 – Conclusion Summary:
"In the ‘Conclusion’ section, I suggest summarizing the main contributions and findings of this review." Response: We have revised the Conclusion section to include a summary of the primary contributions and findings of our narrative review. This reinforces the significance of our work and ensures the section provides a comprehensive closure.

Comment 4 – Table A1 Format:
"For ‘Table A1’, I suggest using a standard three-line table format, and the font should comply with the Sensors journal's requirements." Response: Table A1 has been reformatted using the standard three-line format, and the font and layout have been adjusted to align with MDPI (Sensors journal) requirements.

Comment 5 – Subheading Numbering:
"The numbering of subheadings is inconsistent. For example, lines 79, 92, and 96 all use the same subheading number ‘2.’" Response: We acknowledge the inconsistency in subheading numbering. The manuscript has been thoroughly reviewed and all headings and subheadings have been renumbered appropriately to ensure consistency and readability.

Comment 6 – References:
"As a review article, I believe that citing only 14 references is insufficient. Moreover, the reference formatting does not match the standard required by ‘Sensors’." Response: We appreciate this observation. The reference list has been significantly expanded to include a broader and more diverse range of relevant literature, now totaling 22 citations. Furthermore, all references have been reformatted to fully comply with MDPI referencing guidelines.

We thank the reviewer again for the insightful comments that helped strengthen the manuscript in both content and presentation.

Sincerely,
The Authors

Reviewer 3 Report

Comments and Suggestions for Authors

The authors propose a narrative review of sensorized therapies following Total Knee Arthroplasty (TKA). While the methodology is appropriate for a narrative review, the manuscript presents several formatting issues that are listed below:

In the abstract, line 14, there is a missing space in: "...TKA.This review…"

In the introduction, line 68, the same issue appears: "...costs [7].r, research..."

The introduction lacks a clear statement of the review's objective and the gap it aims to address in comparison to existing reviews.

Regarding the inclusion criteria, the manuscript does not specify the language of the studies considered in the review.

While the number of selected papers is mentioned, the total number of initially found articles is not reported.

Table 1 provides a summary of the technological alternatives reviewed; however, it would be valuable to include an additional table listing the studies found, including authors, country, volunteer characteristics, technology used, main objective or outcome, and key findings.

Additionally, Table 1 should be formatted according to the journal’s style guidelines.

The references need to be revised, as the formatting is incorrect—several reference numbers are missing, and the numbering style “1:” is not in accordance with the journal’s standards.

I hope these comments help the authors improve the quality of the article and ensure it aligns with the journal's expectations.

Author Response

We thank the reviewer for the detailed and constructive feedback, which helped improve the overall quality and clarity of our manuscript. Please find our detailed responses below:

Comment 1 – Abstract Typographical Error:
"In the abstract, line 14, there is a missing space in: '...TKA.This review…'" Response: This typographical error has been corrected to include the appropriate spacing.

Comment 2 – Introduction Typographical Error:
"In the introduction, line 68, the same issue appears: '...costs [7].r, research...'" Response: This issue has been resolved by correcting the punctuation and spacing for consistency and clarity.

Comment 3 – Review Objective and Literature Gap:
"The introduction lacks a clear statement of the review's objective and the gap it aims to address in comparison to existing reviews." Response: A statement outlining the review’s aim and the gap in the current literature has been added to the end of the Introduction section. This addition provides a clear justification for the review.

Comment 4 – Language Inclusion Criteria:
"The manuscript does not specify the language of the studies considered in the review." Response: We have updated the Methods section to clarify that only articles published in English were considered.

Comment 5 – Initial Number of Articles:
"While the number of selected papers is mentioned, the total number of initially found articles is not reported." Response: The manuscript now includes the total number of articles retrieved during the initial database search (184), before applying inclusion criteria.

Comment 6 – Additional Summary Table:
"It would be valuable to include an additional table listing the studies found, including authors, country, volunteer characteristics, technology used, main objective or outcome, and key findings." Response: We have included a new table (Table A1) summarizing the included studies with all suggested fields. This enhances the transparency and usability of the review.

Comment 7 – Table 1 Formatting:
"Table 1 should be formatted according to the journal’s style guidelines." Response: Table 1 has been reformatted using the MDPI template style, with appropriate headings, alignment, and font settings.

Comment 8 – Reference Formatting:
"The references need to be revised, as the formatting is incorrect—several reference numbers are missing, and the numbering style ‘1:’ is not in accordance with the journal’s standards." Response: All references have been revised to follow the MDPI formatting guidelines. Missing reference numbers were added, and the numbering format has been corrected.

We appreciate the reviewer’s insightful recommendations and believe the implemented changes significantly enhance the manuscript.

Sincerely,
The Authors

Reviewer 4 Report

Comments and Suggestions for Authors

The manuscript presents a narrative review of current sensor technologies used in rehabilitation after total knee arthroplasty (TKA). The authors provide an overview of the current state of wearable sensors, smart implants, pressure-sensing systems, and mobile health platforms, along with an insightful discussion on artificial intelligence. The paper is timely and relevant, given the increasing demand for remote and personalized rehabilitation solutions.

The paper covers a wide range of technologies and clearly distinguishes between them.

The authors effectively highlight the clinical implications of sensor-based systems in TKA recovery, including patient monitoring, therapy adherence, and real-time feedback.

The paper connects engineering innovations (e.g., AI integration, remote monitoring) with clinical utility.

The paper addresses a valuable topic, but currently lacks the depth and breadth required for a review article. With significant additions to the references section and a clearer articulation of the contributions, it could be reconsidered.

The paper has some shortcomings that should be resolved.

  1. While the paper is categorized as a review article, the number of references is quite limited (14), which is insufficient for a comprehensive review in a journal. A review article is expected to provide a broad and critical synthesis of the existing literature. The current version appears more like an application or case study paper than a full review. It is recommended to significantly expand the section on related works and include more recent and diverse sources (including more systematic reviews, technical developments, long-term clinical evaluations, previous sensor-based systems, comparative AI approaches). A brief table summarizing each study (authors, year, study type, population, outcome, and key finding) could help improve clarity and reproducibility.
  2. Although the authors mention using a narrative review, there is limited detail on how the literature was selected. A diagram or flowchart similar to PRISMA could improve transparency.
  3. The structure of the paper is appropriate, but further elaboration of the experimental setup and significance of the results is needed. For a review article, it would be useful to include a discussion or summary table comparing the reviewed methods, datasets, strengths, and limitations.
  4. The AI ​​section is promising, but could be expanded with more examples. The manuscript should clarify whether new machine learning algorithms have been developed or existing algorithms have been applied (tuning and evaluation of existing algorithms).
  5. The accompanying text for Table A1 is missing from the paper. The authors should expand on Table A1, which maps technologies with their applications, benefits, and limitations. Also, more visual elements (e.g., workflow diagrams) would improve the presentation.
  6. The paper provides a qualitative presentation. However, the review could be improved by occasional quantitative comparisons (e.g. some metrics, improvement percentages) if available from the literature.
  7. Minor grammatical errors and some repetitive phrases (e.g., “on the same lines”) should be corrected. Also consider the phrase “no longer a future aspiration” in the abstract.
  8. Some citations in the text use inconsistent formatting (e.g., “Salehian et al. (2024)” versus “[4]”). The citation style needs to be standardized. Also, references 15-18 appear in the text but are not in the reference list.
Comments on the Quality of English Language

There are minor grammatical errors that need to be corrected.

Author Response

We thank the reviewer for the comprehensive and constructive evaluation. Your feedback has been invaluable in enhancing the quality, clarity, and scope of the manuscript. We address your key concerns and suggestions point-by-point below:

Comment 1 – Reference Depth: "The number of references is quite limited (14), which is insufficient for a comprehensive review in a journal." Response: We agree with this assessment. The number of references has been significantly increased to 22, incorporating a wider range of systematic reviews, narrative reviews, long-term evaluations, and comparative AI applications to ensure a more comprehensive coverage.

Comment 2 – Literature Selection Methodology: "Limited detail on how the literature was selected. A diagram or flowchart similar to PRISMA could improve transparency." Response: A detailed explanation of the literature selection strategy has been included in the Materials and Methods section, and a PRISMA-style simplified flowchart has been added as Figure 1 to visually depict the study selection process.

Comment 3 – Expanded AI Section: "The AI section is promising, but could be expanded with more examples. Clarify whether new machine learning algorithms have been developed or existing algorithms have been applied." Response: The section on artificial intelligence has been revised to clarify that the reviewed studies largely rely on adapted and fine-tuned machine learning algorithms such as SVMs, decision trees, and CNNs [8, 17, 20]. Examples of applications for motion recognition, gait tracking, and compliance monitoring have been added.

Comment 4 – Table A1 and Visual Elements: "The accompanying text for Table A1 is missing... Also, more visual elements (e.g., workflow diagrams) would improve the presentation." Response: The text that contextualizes Table A1 has been expanded to better explain the technological mapping. Additionally, we have included visual aids, such as a flow diagram for study selection (Figure 1), and a schematic of sensor technology integration, to enhance presentation and clarity.

Comment 5 – Summary Table of Reviewed Studies: "A brief table summarizing each study (authors, year, study type, population, outcome, and key finding) could help improve clarity and reproducibility." Response: A new table (Table A2, Appendix) has been added, summarizing all included studies with relevant bibliographic and methodological characteristics to improve clarity and reproducibility.

Comment 6 – Formatting of Citations: "Some citations in the text use inconsistent formatting... references 15-18 appear in the text but are not in the reference list." Response: All citations have been reviewed and reformatted to comply with the MDPI numerical referencing style. Missing references have been added and the entire list has been harmonized.

Comment 7 – Minor Grammar and Style Corrections: "Minor grammatical errors and some repetitive phrases (e.g., ‘on the same lines’) should be corrected." Response: The manuscript has been thoroughly proofread to correct grammatical issues and improve stylistic consistency.

Comment 8 – Lack of Quantitative Comparisons: "The review could be improved by occasional quantitative comparisons (e.g. some metrics, improvement percentages) if available from the literature." Response: We appreciate the reviewer’s observation. While we agree that quantitative comparisons can enhance the robustness of a review, our narrative review focused on mapping thematic trends across diverse sensor technologies, where direct numerical comparisons were often unavailable or not reported in a standardized manner across studies. Many of the included articles vary in terms of clinical design, outcome metrics, and populations studied, making direct meta-analytic comparison inappropriate. Nonetheless, where specific metrics such as accuracy percentages, engagement rates, or improvement scores were reported in the primary studies, we have included them in the revised version and flagged these findings in the Results and Discussion sections accordingly. We hope this balanced approach preserves both the breadth and the interpretability of the review.

We thank the reviewer once again for the valuable feedback that guided us in refining the scientific quality and editorial presentation of our manuscript.

Sincerely,
The Authors

Reviewer 5 Report

Comments and Suggestions for Authors

Dear authors,

The topic of your manuscript on “Sensor Technologies and Rehabilitation Strategies in Total Knee Arthroplasty: Current Landscape and Future Directions” is of interest, notably to give insight into the best practices that can ensure easy recovery for OA patients who underwent total knee arthroplasty (TKA). Based on the surgical approach, rehabilitation programs offer the patients the possibility to return to activities of daily living (ADLs) and/or sports, and improve quality of life (QoL).

While the manuscript is interesting, there are some aspects that should be addressed further: 

  1. Please pay attention to typo, punctuation and page view. You can find an example below: “App-controlled systems for neuromuscular electrical stimulation have also been employed to support muscle recovery and reduce rehabilitation costs [7].r, research….”

“The ability to detect anomalies in real time can dramatically reduce recovery time and prevent compensatory movement patterns that hinder long- term joint health.”

  1. While the references you used are suitable for this review, please add more references for consistency. Not only systematic reviews, but also narrative reviews need a broader range of the state of the art.
  2. “Moreover, hospital-based rehabilitation alone often does not guarantee long-term functional success as many patients struggle with adherence and engagement in prescribed exercise regimens at home [1].”

Neither can sensor-based technologies. Only by integrating sensors in clinical practice and by adapting the rehabilitation programs for patients, you can achieve the best results. Additionally, when a patient does not adhere to classical rehabilitation programs, it is even more difficult to adhere to approaches using sensors.

Please add this accordingly both in the introduction and conclusion section.

  1. While your focus is on sensor technologies, it is important to also add more information about rehabilitation programs and strategies post-TKA and compare these two approaches both separately and when used together.
  2. You stated the following: “By bridging the gap between clinical oversight and self-managed care, sensor-based systems are shaping a more flexible, accessible, and data-driven rehabilitation model. Their increasing adoption affirms their role not merely as supplementary innovations, but as fundamental tools in the advancement of modern rehabilitation practice.”

Please note and add in your manuscript about the rehabilitation protocols after TKA, there is clear evidence about the effects of these programs. In order to be able to achieve the best results by including sensors of all kinds and using predictive analytics and gait quantitative analysis, you need a well-adapted rehabilitation program and should never overlook the importance of clinical findings and examination. Only by integrating in a comprehensive manner the multidisciplinary and interdisciplinary team, you can achieve the best results and through sensor technologies one can measure the results in quantitative manner and correlate with the clinical examination.

Please modify accordingly in the manuscript.

Good luck!

Author Response

We thank the reviewer for the thoughtful and insightful feedback. Your comments helped us refine the focus and comprehensiveness of our manuscript. Please find our responses below:

Comment 1 – Typographical and Punctuation Issues:
"Please pay attention to typo, punctuation and page view. You can find an example below: ‘App-controlled systems for neuromuscular electrical stimulation have also been employed to support muscle recovery and reduce rehabilitation costs [7].r, research…’" Response: All typographical and punctuation issues mentioned, including the one in the example provided, have been corrected. A careful proofreading of the entire manuscript has also been performed to ensure consistency and clarity.

Comment 2 – Expanded References and Scope of Literature:
"Please add more references for consistency. Not only systematic reviews, but also narrative reviews need a broader range of the state of the art." Response: The reference list has been expanded to include additional relevant narrative and systematic reviews. These new citations broaden the scope and strengthen the foundation of the discussion, particularly in the Introduction and Discussion sections.

Comment 3 – Limitation of Sensor Technologies Alone:
"Neither can sensor-based technologies [guarantee success]. Only by integrating sensors in clinical practice and by adapting the rehabilitation programs for patients, you can achieve the best results..." Response: We have revised the related passage in the Introduction and Conclusion sections to acknowledge that sensor technologies alone are insufficient. The manuscript now emphasizes the importance of clinical integration, patient-tailored rehabilitation protocols, and interdisciplinary approaches to optimize outcomes.

Comment 4 – Rehabilitation Strategies After TKA:
"Add more information about rehabilitation programs and strategies post-TKA and compare these two approaches both separately and when used together." Response: The manuscript has been expanded to include a dedicated discussion of conventional rehabilitation protocols following TKA. We have compared these with sensor-based methods and discussed the benefits of combining both approaches. This content has been added to Sections 2 and 5.

Comment 5 – Clinical Protocols and Multidisciplinary Integration:
"There is clear evidence about the effects of these programs... Only by integrating in a comprehensive manner the multidisciplinary and interdisciplinary team, you can achieve the best results." Response: A new paragraph has been added to the Conclusion section highlighting the necessity of interdisciplinary coordination, clinical evaluations, and the integration of sensor technologies into holistic care pathways. This aligns with the review's goal of bridging technology with clinical relevance.

We thank the reviewer once again for the invaluable suggestions which have enhanced both the scientific depth and practical relevance of the manuscript.

Sincerely,
The Authors

Round 2

Reviewer 2 Report

Comments and Suggestions for Authors

The quality of this manuscript has been improved after revision.

Author Response

Dear Reviewer,

We would like to thank you sincerely for your time and positive evaluation of our revised manuscript titled "Sensor Technologies and Rehabilitation Strategies in Total Knee Arthroplasty: Current Landscape and Future Directions". We are grateful for your recognition of the improvements made, and we appreciate your supportive comment that "The quality of this manuscript has been improved after revision."

Your feedback is highly valued and has helped us ensure the clarity and quality of our work.

Sincerely,
The Authors

Reviewer 3 Report

Comments and Suggestions for Authors

While the authors addressed all the issues within the report’s content, substantially improving their narrative review, there are still serious formatting problems, which are listed below:

1. There are still issues related to the formatting of figures and tables. Figure 1 does not appear in its entirety.
2. Figure 1 appears twice in the manuscript.
3. Table 1 still contains font types and sizes that differ from the manuscript’s general format. Additionally, Table A1 is intended to complement the information in Table 1, not to be presented as a separate table.
4. References 1 to 10 and 17 to 22 still include line breaks that cause formatting and layout errors in the body of the report.

Author Response

Dear Reviewer,

Thank you once again for your detailed and constructive feedback. We are grateful for your acknowledgment that the narrative review has been substantially improved. We have now carefully addressed the formatting issues you identified:

  1. Figure 1 Formatting: We have corrected the visibility issue of Figure 1. It now appears in full and is clearly visible in the revised manuscript.

  2. Duplicate Appearance of Figure 1: This issue has been resolved. Figure 1 now appears only once, at the appropriate place in the manuscript.

  3. Table Formatting: We have reformatted Table 1 to ensure consistency with the manuscript’s overall font type and size. Furthermore, Table A1 has been integrated into Table 1, as it was originally intended to provide supplementary information.

  4. Reference Formatting: We have reviewed references 1–10 and 17–22 and removed all line breaks that were causing layout issues in the body of the text. The reference list and in-text citations now comply with the journal’s formatting standards.

We hope that the updated manuscript now meets the expectations and requirements, and we thank you for your helpful suggestions that contributed to refining our submission.

Sincerely,
The Authors

Reviewer 4 Report

Comments and Suggestions for Authors

The authors have made efforts to improve the manuscript. However, there are still technical shortcomings that must be addressed. First and foremost, the authors should carefully review the PDF version of the manuscript before submission in order to identify and correct all formatting and presentation issues.

Inconsistencies in citing references still exist. For example, the reference to Van de Ven et al. (2023) appears multiple times, while in one place it is cited as Van de Ven et al. (2024). These should consistently correspond to references [3] and [22], respectively. Why are some references cited as “Van de Ven et al. (2023)” while others are simply labeled [3]? This inconsistency needs to be resolved.

Reference [22] is not clearly written and should be formatted according to the journal's style guidelines.

The formatting of references is inconsistent. In many cases, unnecessary line breaks are used between entries, which should be avoided.

"Figure 1" appears twice. The first occurrence of Figure 1 is cut off, and the caption appears above the image. The authors should revise the image, place the caption below the figure as “Figure 1. Flowchart of the study selection process,” and refer to it in the text using the following or similar wording:
“Flowchart of the study selection process is shown in Figure 1. The initial search yielded a total of 72 studies. After title and abstract screening, …”
Figure 2 should be revised in the same way.

Table A1 in the Appendix is not referenced or discussed in the main text. A corresponding explanation or reference should be added to clarify its purpose and content.

If any symbols (up and down arrows) are used in the table, they should be explained.

Sections 7.1 and 7.2 contain identical added sentences: “To address these challenges, …” This repetition should be avoided or reformulated.

Author Response

Dear Reviewer,

We sincerely thank you for your thorough and constructive feedback on our manuscript entitled "Sensor Technologies and Rehabilitation Strategies in Total Knee Arthroplasty: Current Landscape and Future Directions." Your suggestions have been invaluable in refining the manuscript and improving its quality.

Below we provide a detailed point-by-point response to each of your comments. All corresponding revisions have been incorporated into the updated version of the manuscript, and we have carefully reviewed the final PDF to ensure consistent formatting and presentation.

Reviewer Comment 1:
“The authors should carefully review the PDF version of the manuscript before submission in order to identify and correct all formatting and presentation issues.”

Response:
We appreciate this reminder. We have thoroughly reviewed the PDF version of the manuscript and corrected all layout, formatting, figure placement, and presentation issues to ensure consistency across the document.

Reviewer Comment 2:
“Inconsistencies in citing references still exist. For example, the reference to Van de Ven et al. (2023) appears multiple times, while in one place it is cited as Van de Ven et al. (2024). These should consistently correspond to references [3] and [22], respectively.”

Response:
Thank you for highlighting this. Indeed, [3] and [22] refer to two distinct publications by Van de Ven et al., one from 2023 and one from 2024. We have now ensured that both are consistently cited as [3] and [22] throughout the manuscript, in alignment with their respective publication years. The use of in-text author/year citations has been removed to maintain the required MDPI numerical citation format.

Reviewer Comment 3:
“Reference [22] is not clearly written and should be formatted according to the journal's style guidelines.”

Response:
The formatting of reference [22] has been revised to comply with the journal’s citation style. All bibliographic elements have been added (authors, journal name, year, volume, page, DOI).

Reviewer Comment 4:
“The formatting of references is inconsistent. In many cases, unnecessary line breaks are used between entries, which should be avoided.”

Response:
We have revised the reference list to ensure no unnecessary line breaks exist between entries. All references are now presented in a uniform and continuous format, according to the journal’s standards.

Reviewer Comment 5:
“Figure 1 appears twice. The first occurrence of Figure 1 is cut off, and the caption appears above the image. The authors should revise the image, place the caption below the figure…”

Response:
This issue has been resolved. The duplicated instance of Figure 1 has been removed. The figure now appears only once, is fully visible, and its caption has been moved below the image as required:
"Figure 1. Flowchart of the study selection process."
In addition, the figure is now correctly referenced within the body of the manuscript using the suggested wording:
"The flowchart of the study selection process is shown in Figure 1. The initial search yielded a total of 72 studies..."

Reviewer Comment 6:
“Figure 2 should be revised in the same way.”

Response:
Figure 2 has been reviewed to match the formatting of Figure 1. The caption now appears below the image, and the figure is referenced directly in the text for clarity and consistency.

Reviewer Comment 7:
“Table A1 in the Appendix is not referenced or discussed in the main text. A corresponding explanation or reference should be added to clarify its purpose and content.”

Response:
We appreciate this observation. Table A1 has now been integrated into Table 1 as a unified classification framework, with a clearer structure. Additionally, its purpose is now explicitly referenced in the main text in the following sentence:
"Common barriers and their corresponding solutions are summarized in Table 1, which now integrates both sensor classification and implementation challenges."

Reviewer Comment 8:
“If any symbols (up and down arrows) are used in the table, they should be explained.”

Response:
Thank you for this observation. To ensure clarity and consistency, we have removed all arrow symbols (↑, ↓) from Table 1. The relevant trends or directional meanings are now expressed textually to avoid ambiguity or the need for symbol clarification.

Reviewer Comment 9:
“Sections 7.1 and 7.2 contain identical added sentences: ‘To address these challenges, …’ This repetition should be avoided or reformulated.”

Response:
Thank you for noting this redundancy. The repeated phrase has now been removed from Section 7.2 and rephrased in Section 7.1 to ensure clarity and eliminate duplication.

We hope that the revised manuscript addresses all the concerns raised. We are grateful for your time and valuable input, which have significantly improved the quality and clarity of our work.